# Exploring the links between social connection and physical functioning among older Adults: A network analysis

**Chuwen Zhong**[1]*, **Linh Dang**[1,2], **Aparna Ananthasubramaniam**[3,4], **Briana Mezuk**[1,2‡]

**1** Department of Epidemiology, University of Michigan School of Public Health, Ann Arbor, Michigan, United States of America, **2** Research Center for Group Dynamics, Institute for Social Research, University of Michigan, Ann Arbor, Michigan, United States of America, **3** School of Information, University of Michigan, Ann Arbor, Michigan, United States of America, **4** School of Social Work, University of Michigan, Ann Arbor, Michigan, United States of America

‡ These authors are joint senior authors on this work.
* zcwyolo@umich.edu

## Abstract

### Purpose

The aims of this study were to 1) characterize and visualize the association between social connection and physical functioning using network analysis, with attention to different types of measures of these constructs, and 2) identify key indicators that link social connection and physical functioning.

### Methods

Data come from the 2014/2016 Health and Retirement Study (N = 7,270, mean age = 67.3). Network analysis was used to depict and explore the relationships between physical functioning and social connection. Both of these constructs were measured using a combination of "objective" (e.g., balance test, number of social connections) and "subjective" (e.g., self-evaluated activities of daily living (ADLs), perceived loneliness) indicators. The network was estimated using a regularized partial correlation network. Features of the network structure were characterized using 1) assortativity, 2) community detection, and 3) centrality indices (i.e., strength and betweenness). Betweenness centrality was used to identify the key nodes linking the two constructs.

### Results

The network structure is determined by constructs (i.e., social connection vs. physical functioning, assortativity = 0.87), instead of types of measures (i.e., objective vs. subjective, assortativity = 0.45). There are five communities identified from the network, three of which include both subjective and objective indicators. The *incomplete*

**Data availability statement:** Data from this study were from the Health and Retirement Study, which is publicly accessible at https://hrs.isr.umich.edu/ This study used the following datasets 1) RAND HRS Longitudinal file 2018 v1 2) 2014 RAND HRS Fat file 3) 2016 RAND HRS Fat file 4) Cross-Wave Tracker File After registering with HRS, these datasets and corresponding codebooks and documentations can be accessed at https://hrsdata.isr.umich.edu/data-products/rand.

**Funding:** This study was financially supported by the National Institute of Mental Health in the form of a grant awarded to BM (R01-MH128198). No additional external funding was received for this study. The funder had no role in study design, data collection and analysis, decision to publish, or preparation of the manuscript.

**Competing interests:** The authors have read the journal's policy and have the following competing interests: LD was supported by a training grant from the National Institute on Aging for her PhD training (T32AG027708). This does not alter our adherence to PLOS ONE policies on sharing data and materials.

*balance test* is the key bridging indicator that links social connection and physical functioning.

## Conclusion

The linkages between social connection and physical functioning are multifaceted. Balance impairment is the key bridge between these constructs for older adults.

## Introduction

Improving physical functioning and social connection are essential for supporting the overall well-being of older adults. Two-thirds of US adults aged 65 + have at least one functional limitation, and 43% of adults aged 60 + report feeling lonely [1,2]. Physical functioning and social connection are often mutually-reinforcing. For example, older adults with functional limitations are more likely to experience reduced space for daily movement and decreased social activities [3]. Greater understanding of the linkages between physical functioning and social connection can inform the development of interventions that support the overall well-being of older adults, including policies promoting an aging-friendly environment. This paper applied psychometric network analysis to explore the complex interactions between physical functioning and social connection, with an emphasis on identifying specific indicators of physical functioning and social connection that strengthen the relationship between these two constructs.

### Physical functioning

Physical functioning is a complex construct, as it involves multiple body systems, such as the cardiovascular, musculoskeletal, and neurological systems, and it is sensitive to changes in the physical environment, such as flooring [4,5]. Moreover, the trajectory of physical functioning decline is not linear; some physical functions may show gradual deterioration during early phases, and then decline abruptly when beyond a certain threshold [6]. Therefore, diverse measurements are needed to capture its full complexity.

This paper focuses on two distinct approaches to measuring physical functioning in population-based research: objective and subjective measures. The *objective* measure refers to standardized tests that are used to assess an individual's physical functioning [7]. For example, balance tests and grip strength are two widely used physical performance assessments for muscle strength and joint stability. In contrast to physical performance tests, which measure seconds taken to walk on timers or the kilograms a hand gripped on dynamometers, the *subjective* measure of physical functioning asks respondents to evaluate whether they have difficulty with specific activities of daily living, such as walking and preparing a meal [5]. This subjective judgement reflects both their actual physical ability as well as perception and evaluation of environmental constraints or adaptations. Although objective and subjective measures are correlated, they do not always agree with each other [8,9]. For example, a person may have weak lower-body strength, but lives in a home that

has an adaptive and accessible bathroom, and therefore, the person might report no difficulty in bathing. Understanding and measuring physical functioning in both subjective and objective measures is important to evaluate and develop target interventions.

### Social connection

Social connection is an umbrella term for how people relate and interact with others [10]. Like physical functioning, the construct of social connection encompasses several components that are correlated but conceptually distinct [2]. This paper focuses on two well-studied components: social isolation and loneliness. Social isolation reflects the *objective* aspect (i.e., structure) of social connection, referring to whether individuals report limited social relationships and infrequent social contact [2]. In contrast, loneliness reflects the *subjective* aspect of social connection, referring to whether individuals feel their existing social connections meet their social needs. Indeed, social isolation and loneliness are positively correlated but distinct concepts; for example, people can report feeling lonely even if they have many social ties [11]. Both isolation and loneliness vary over the life course. For example, the overall number of social ties typically declines as people age [12], and individuals who become widowed are more likely to experience loneliness [13].

### Associations between social connection and physical functioning

A well-established body of research has documented a bidirectional relationship between physical functioning and social connection among older adults. Social isolation and loneliness are significant predictors of many physical health outcomes over the life course, including frailty, activity levels, and physical functioning limitations [11,14,15]. Conversely, physical impairment often leads to decreased engagement in social activities, which in turn may also accelerate the development of functional limitations [16]. These findings illustrate the complex interactions between physical functioning and social connection. This complexity poses a challenge in disentangling this association to inform targeted interventions for improving physical function and promoting social connection among old adults.

The Disablement Process Model (DPM) provides a useful framework for characterizing the relationship between physical functioning and social connection [17]. Specifically, the DPM emphasizes that physical impairment (i.e., declines in objective measures of physical functioning like musculoskeletal strength) does not necessarily lead to disability (i.e., subjective experiences of difficulty in performing activities of daily living); rather, this disablement process is shaped by social, psychological, and environmental factors like loneliness and social isolation [17,18]. For instance, older adults with poor balance may have fear of leaving the house, so they lose contact with friends. Further, less contact with friends may lead to feelings of loneliness, reducing motivation for activities like cleaning. In this example, designing effective interventions to address loneliness would need to consider physical functioning and its relationship to social connection (e.g., poor balance leading to less contact with friends) as well as the relationship between the objective and subjective symptoms of social connection (e.g., less contact with friends leading to feelings lonely). This underscores the need to *simultaneously* consider how multiple indicators of social connection and physical functioning are interrelated with each other (e.g., poor balance, less contact with friends, isolation, and difficulty cleaning), rather than treating each indicator separately, or targeting at an aggregated level.

### Current study

Investigating the relationships between multiple indicators of social connection and physical functioning via conventional modeling presents two methodological limitations. Firstly, conventional analysis typically relies on aggregated indices (e.g., mean scores of a loneliness scale), which assume that all individual indicators (i.e., questionnaire items) are equally important to the construct. For instance, when assessing loneliness using the UCLA loneliness scale, we assume that feeling left out is similarly essential to feeling not understood by others. This assumption may not be accurate [19], and it



masks the opportunity to identify which specific indicators drive the association between constructs. Second, while regression analysis can examine how individual indicators of social connection (as exposures) relate to a particular indicator of physical functioning (as outcome) or vice versa, it cannot *simultaneously* show how all individual indicators of social connection and physical functioning correlate within an interrelated structure. Understanding the structure of the interrelated relationship can inform interventions to disrupt the vicious cycle between poor social connection and poor physical functioning, or to improve them together.

To address the need for a systemic approach to advance understanding of social connection and physical functioning, the present study employed network analysis. Network analysis can visualize and quantify the structure and patterns of interrelationship among multiple constructs at the indicator level, and it has been widely applied in clinical psychology to investigate the association between two mental health disorders and to identify symptoms driving comorbidity [20–22]. This study extended the application of network analysis to explore the relationship between physical functioning and social connection. Specifically, this study aimed to examine [1] how robustly indicators of physical functioning and social connection are interrelated, [2] the extent to which each individual indicator drives the relationship between social connection and physical functioning, [3] how strongly objective indicators and subjective indicators are connected, and [4] which sets of indicators strongly cluster together.

## Materials and methods

### Sample

The Health and Retirement Study (HRS) is a nationally representative longitudinal survey of US adults age 50 and older, with approximately 20,000 respondents, conducted biennially [23]. Beginning in 2006, the HRS cohort was split into random 50% subsamples, each of which completed Enhanced Face-to-Face interviews (EFTF) in alternating waves. The EFTF interview collected biological and physical data, including physical performance measures. Respondents were then asked to complete the self-administered Psychosocial Leave-Behind Questionnaire (PLBQ), which included measures of loneliness and social network characteristics. Additional details of the HRS have been described elsewhere [23]. The present study is a cross-sectional analysis using a pooled sample of respondents who completed the EFTF interview in either 2014 or 2016 and had complete data on physical performance tests, measures of social connection, and demographic covariates (N = 7,270). The analytical sample selection process is detailed in S1 Fig in S1 Table. Because the network analysis only allows complete cases, the analytic sample excluded respondents with missing values in any of the 34 nodes. Excluded respondents were more likely to be female, to be racial/ethnic minorities, and to have lower socioeconomic status (**S1 Table**).

The HRS is approved by the Institutional Review Board at the University of Michigan and all respondents are provided written informed consent. This analysis used only publicly available data and was exempt from human subjects regulation. Data is available at https://hrs.isr.umich.edu/.

### Measures

Our analysis included a total of 34 indicators, including 14 indicators for physical functioning (2 objective, 12 subjective) and 20 indicators for social connection (9 objective, 11 subjective). All indicators were dichotomized (e.g., yes/no, high/low), as detailed in S2 Table in S1 Table. All indicators were weak to moderately correlated (coefficients ranged from −0.04 to 0.6, see S2 Fig in S1 Table). Mean and standard deviation of each indicator are reported in S4 Table in S1 Table.

**Physical functioning.** Subjective indicators encompassed self-reported activities of daily living (ADLs) and instrumental activities of daily living (IADLs). Interviewers asked respondents: "Because of a health or memory problem, do you have any difficulty with a certain activity?" in relation to six ADLs (i.e., bathing/showering, dressing, walking, getting in/out of bed, using the toilet, and eating) and six IADLs (i.e., using a map, using a cellphone, managing money, shopping,

taking medication, and preparing hot meals) [24]. Response options included "yes", "no", "can't do", or "don't do". Both ADL and IADL had high reliability (Cronbach's alpha = 0.85 for ADL, and 0.86 for IADL). We dichotomized each activity into "no difficulty" vs. "any difficulty".

The two performance-based measures included the grip strength and the balance test. Grip strength (in kilograms) was measured for both hands [25]. This analysis used the maximum grip strength reported for the dominant hand, and it was further dichotomized at the median (i.e., 29 kg) into low and high. For the balance test, respondents were asked to hold a semi-tandem stance for 10 seconds. For those who passed, they were additionally asked to hold a full-tandem stance for 10 seconds or 60 seconds, depending on the age of the respondent [25]. The *incomplete balance tests* was dichotomized by whether the respondent completed both the semi-tandem and full tandem tests (complete both vs. incomplete either). Respondents who could not complete both balance tests had balance impairment. Both tests yielded good intra-rater reliability (Intraclass correlation coefficient, ICC > 0.7) [26,27].

**Social connection.** We used eleven items derived from the Revised UCLA loneliness scale (R-UCLA) to measure loneliness [28,29]. The scale is significantly correlated with other measures of loneliness and has high reliability (Cronbach's alpha = 0.87) in assessing loneliness in older adults [28]. It is assessed by asking respondents how often they endorsed feelings on four items of emotional loneliness and seven of social loneliness. The answers were on a 3-point Likert scale (1 = often, 2 = some of the time, 3 = hardly ever or never), though this analysis dichotomized responses into "high" and "low" levels of loneliness. "High" was defined as reporting "often" or "some of the time" on negatively worded items endorsing feelings of loneliness (e.g., feeling alone) or "hardly ever" or "never" on positively worded items(e.g., feeling in tune with others).

We measured social isolation using the Social Disconnectedness Scale, which has high reliability (Cronbach's alpha = 0.73) [30]. Four indicators of social network diversity measured whether the respondent had a spouse/partner, living children, other immediate family members, and friends. Frequency of contact, either in-person, by phone, writing/emailing, or social media, was assessed separately, summed, and dichotomized into low frequency of contact for each type of relationship, based on the median. Not feeling close to their spouse/partner was defined as reporting "not at all close" or "not very close" with their spouse/partner. Infrequent social participation was defined as participating in at least one of the eight social activities from the Social Engagement Scale (e.g., training class, volunteering) less than several times a month.

**Covariates.** The relationships between the 34 indicators listed above were adjusted for demographic covariates that, based on prior work, were determined to be associated with both physical functioning and social connection. These included age (grouped into 5-year intervals from 51 to 80+), sex (male, female), race/ethnicity (non-Hispanic White, non-Hispanic Black, Hispanic, and Other), education attainment (less than high school, high school or equivalent, some college, college or above), household income (in quartiles), and total wealth (in quartiles).

## Analysis

**Generating the network.** Psychometric networks are a descriptive tool used to graphically depict and characterize the complex relationships between different constructs [22]. A network consists of entities (called *nodes*) and the relationships between these entities are depicted by lines (called *edges*). In our network, nodes are individual indicators of two constructs (physical functioning and social connection), and edges represent the associations between pairs of these indicators. Each edge's *sign* (>0 or <0) and *weight* (ranging from 0 to infinity) indicate, respectively, the orientation (positive or negative) and strength of the corresponding association [31].

The network was estimated using the Least Absolute Shrinkage and Selection Operator (LASSO) with extended Bayesian information criterion (EBIC) to identify edges, so-called eLasso [32]. This was done because a network constructed from all possible partial correlations between 34 nodes was extremely dense and visually uninformative. The application of eLasso can result in a sparse network by shrinking the weight of weak edges to exactly zero, which effectively removes

those edges from the network. The amount of network "shrinkage" is controlled by the EBIC penalty parameter γ. The higher the γ is, the more edges would be shrunk, leading to a sparser network. We set the γ to 0.25 as previous literature shows that models consistently perform the best with parameter γ = 0.25 [32]. eLasso was implemented using the "and-rule", in which edge *a–b* remained in the network only if both $a \rightarrow b$ and $a \leftarrow b$ associations were non-zero [32]. Pairs of nodes with tautological associations based on survey skip logic were not connected in the final network (e.g., respondents who reported no children were not asked about frequency of contact with their children; therefore, no edges were drawn between the "no children" and "low contact with children"). The network accounted for the demographic covariates listed above and the HRS Leave-Behind sampling.

We quantified three key features of the network. First, we calculated the assortativity for measures and constructs. This metric allowed us to examine whether the measures (objective vs. subjective) or constructs (physical functioning vs. social connection) primarily shaped the structure of relationships. The assortativity index quantifies the conceptual coherence of a group, ranging from −1–1 [33]. For example, an assortativity value of 0.9 for constructs and 0.3 for measures indicates that the nodes are grouped by constructs. In other words, a subjective indicator of social connection is more likely to have edges with other indicators of social connection than subjective indicators of physical functioning. Second, we identified the subgroups within the network. This metric allowed us to understand how individual indicators cluster in the network without pre-determined labels (such as constructs or measure types). We used the spin-glass community detection algorithm [31]. The algorithm detected communities by grouping nodes that had more edges within the group and fewer edges between groups [31]. Third, we calculated strength and betweenness centrality to assess the relative importance of nodes, providing a more nuanced and comprehensive understanding of each of them, which can ultimately inform implications. Strength is calculated as the sum of the weights directly connected to a node; indicators with high strength have greater overall influence, meaning that they may present as focal intervention targets to improve overall physical functioning/social connection status [34]. In contrast, *betweenness centrality* is the number of times that a node lies on the shortest path connecting two other nodes; a node with high betweenness tends to bridge other parts of the network [34]. The "bridge" node can be an intervention target to disrupt the vicious cycle between poor social connection and poor physical functioning.

**Sensitivity analyses.** We conducted several sensitivity analyses to assess the robustness of our results. First, to investigate whether the observed network structure could have arisen by chance, the characteristics of the observed network were compared against the average characteristics of 100 randomly generated reference networks [31]. We chose the *configuration* null model for the comparison, which keeps the number of edges connected to each node from the observed network, before drawing edges at random.

Second, to more robustly assess the impact of certain nodes in the network, we employed an ablation test [35]. In this test, we removed a node with high centrality and re-estimated the network; the impact of removing a particular node on the overall structure of the network was quantified by comparing assortativity from the original and ablated network. If the target node plays a significant role in bridging physical functioning and social connection, its ablation should decouple the two constructs, increasing assortativity.

Finally, we tested the stability of results under different values of the penalty parameter (*γ*) used in EBIC by generating networks using a range of γ values between 0 and 1 in intervals of 0.05. For each generated network, we compared the difference in assortativity of the original (γ = 0.25) and alternative network, the fraction of nodes remaining in the same community, and the Spearman correlation of strength and betweenness values across the two networks.

**Statistical testing.** Bootstrapping was used to generate the 95% confidence intervals for assortativity and centrality values, and p-values for comparisons [36]; for each comparison, 10,000 bootstrap networks were generated by resampling the HRS survey data, accounting for each respondent's sampling weight, along with the corresponding ablated and reference networks. From these bootstrap distributions, one-tailed p-values were generated to test if node centrality in the observed network was higher than the null network or the assortativity in the observed network was lower than the

ablated networks, and the corresponding 95% confidence interval was generated for each measure. All p-values and confidence intervals were adjusted for multiple comparisons using a Bonferroni correction.

Analyses were performed in R (v.2023.06.2 + 561). R packages used are listed in S3 Table in S1 Table. Artificial intelligence was not used in the production of this submitted work.

## Results

### Descriptive characteristics

The analytic sample had a mean age of 67.3 years. As shown in **Table 1**, the majority were women (51.83%) and non-Hispanic White (81.20%). The mean R-UCLA score was 2.3. Among the 4 types of social relationships (children, partner/spouse, friends, and other family members), respondents had three types of relationships on average. Less than 10% of respondents had any ADL or IADL limitations. Mean standing time was 54 seconds, and mean grip strength was 33 kg.

### Network structure

**Fig 1** illustrates the psychometric network for the associations between physical functioning and social connection. Each node represents one individual question or test from two types of measures (objective and subjective) of each of the two constructs (social connection and physical functioning). The color of the node is determined by the construct: blue refers to social connection, and yellow refers to physical functioning. The shape of the node is determined by the type of measure: circles refer to subjective measures and boxes refer to objective measures. Each edge represents the association between two nodes while adjusting for other nodes. The thickness and color of each edge are proportional to the strength and direction of the association, where stronger edges are shown in wider and darker lines.

The network is highly clustered by constructs, as blue nodes cluster together and yellow nodes cluster on the other side of the network. This visual interpretation is confirmed quantitatively by the assortativity index (0.87, 95% CI = [0.85,0.87]). The notion that nodes are being clustered by color rather than shape means that the network structure is primarily driven by the construct (physical functioning/social connection), rather than the type of measure (subjective/objective). Objective and subjective measures are more closely connected than separated, as demonstrated by a relatively low assortativity(0.46, 95% CI = [0.43, 0.46]).

**Fig 2** shows the result of the community detection algorithm, illustrating how nodes cluster together without pre-defined labels. The algorithm identifies five communities, which roughly correspond to the subgroups of the two constructs: emotional loneliness (e.g., feeling left out, not having a spouse/partner), social loneliness (e.g., not feeling understood by others, not having any friends), low social contact (e.g., infrequent contact with family, children, and friends), core physical functioning (e.g., ADL, incomplete balance tests), and complex physical functioning (e.g., IADL). All communities have nodes only from a single construct (social connection/physical functioning). However, three of the five communities contain nodes from both subjective and objective measures. For example, the green community includes only physical functioning nodes; however, it has the *incomplete balance tests* (objective) and *the having difficulty with bathing* (subjective). This indicates that although nodes can be classified by subjective and objective measures, their specific content, rather than how they are measured, determines their community membership.

### Nodes importance

**Fig 3a** shows the betweenness centrality of each node, relative to the null network. Specifically, the nodes representing *feeling isolated* (betweenness = 0.27 [0.24,0.27], p = 0.003), *not part of any group* (betweenness = 0.22 [0.21,0.23], p = 0.003), and the *incomplete balance tests* (betweenness = 0.16 [0.16,0.17], p = 0.003) have the highest betweenness values. These three nodes might be important in linking social connection and physical functioning. The ablation test shows that the associations between physical functioning and social connection significantly decrease only after removing

**Table 1. Demographic characteristics of the analytic sample: health and retirement study, 2014/2016 (N = 7,270).**

| Characteristics | Analytic sample (N = 7,270) |
|---|---|
| **Age group, N (weighted %)** | |
| 51–55 yr. | 732 (14.74) |
| 56–60 yr. | 1,447 (24.31) |
| 61–65 yr. | 1,374 (20.97) |
| 66–70 yr. | 1,010 (16.06) |
| 71–75 yr. | 1,050 (10.82) |
| 76–80 yr. | 899 (6.78) |
| 80 + yr. | 758 (6.51) |
| **Sex, N (weighted %)** | |
| Female | 4,160 (51.83) |
| **Race/Ethnicity, N (weighted %)** | |
| Hispanic | 710 (7.47) |
| Non-Hispanic White | 5,265 (81.20) |
| Non-Hispanic Black | 1,032 (7.48) |
| Others | 263 (3.86) |
| **Education, N (weighted %)** | |
| Less than high school | 734 (7.64) |
| High school degree or GED | 2,394 (30.22) |
| Some college | 1,989 (28.28) |
| College or more | 2,153 (33.85) |
| **Marital status, N (weighted %)** | |
| Partnered | 4,745 (68.29) |
| **Household income (in USD), N (weighted %)** | |
| ≤ $22,760 | 1453 (15.72) |
| $22,761−40,992 | 1456 (16.05) |
| $40,993−66,130 | 1453 (18.98) |
| $66,131−111,612 | 1454 (22.31) |
| > $111,612 | 1454 (26.94) |
| **Total wealth (in USD), N (weighted %)** | |
| ≤ $28,950 | 1454 (17.68) |
| $ 28,951-$135,250 | 1454 (19.28) |
| $135,251-$323,200 | 1455 (19.50) |
| $323,201-$752,200 | 1453 (20.94) |
| > $752,200 | 1454 (22.59) |
| **>1 difficulty in ADL, N (weighted %)** | 719 (9.01) |
| **>1 difficulty in IADL, N (weighted %)** | 650 (7.55) |
| **UCLA – loneliness scale, mean (SE)** | 2.27 (0.10) |
| **Social Network Diversity, mean (SE)** | 3 (0.01) |
| **Standing time (s), mean (SE)** | 54 (0.47) |
| **Grip strength (kg), mean (SE)** | 32.96 (0.20) |

Note. SE = standard error. ADL = activities of daily living. IADL = instrumental activities of daily living. Values account for the PLBQ sampling weight and percentages may not sum up to 100% due to rounding.

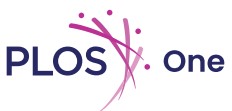

**Fig 1. Network structure of the physical functioning and social connection in the health and retirement study.** Caption for Fig 1: Blue nodes refer to social connection, and yellow nodes refer to physical functioning. Circles refer to subjective indicators (Is: Social connection/subjective, Fs: Physical functioning/subjective) and squares to objective indicators (Io: Social connection/objective, Fo: Physical functioning/objective). See S2 Table in S1 Table for node definitions and operationalizations.

the *incomplete balance tests*, indicating that the *incomplete balance tests* is the most important bridge between physical functioning and social connection in the network. The results of the ablation test are detailed in S3 Fig in S1 Table.

**Fig 3b** shows the strength centrality of each node. Relative to the null network, a handful of nodes has higher strength centrality, including *difficulty with shopping* (strength = 9.98 [9.88,10.06], p = 0.003), *difficulty with taking medication* (strength = 8.03 [7.99,8.16], p = 0.003), *difficulty with dressing* (strength = 7.63 [7.52,7.89], p = 0.003), *feeling not understood* (strength = 6.74 [6.69,6.75], p = 0.003), and *having no one to turn to* (strength = 6.53 [6.45,6.60], p = 0.003). This ranking pattern indicates that the association between physical function and social connection is primarily driven by physical functioning.

Sensitivity analyses showed that changing γ did not substantially change the network's assortativity, communities, or node centrality (S4-6 Fig in S1 Table), meaning different values of γ generated relatively stable networks. Confidence intervals are presented in S5 Table and S7 Fig in S1 Table.

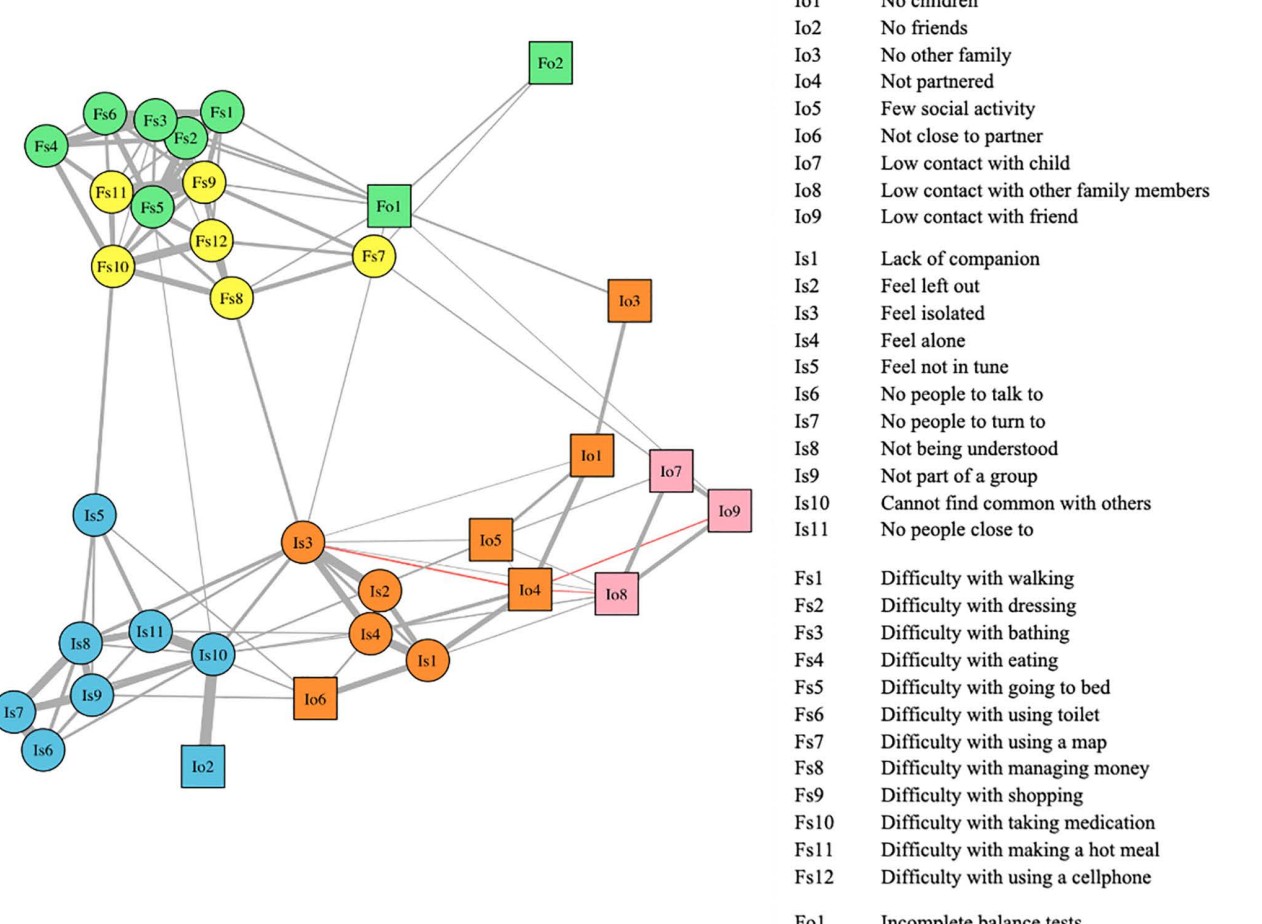

| Label | Key |
| --- | --- |
| Io1 | No children |
| Io2 | No friends |
| Io3 | No other family |
| Io4 | Not partnered |
| Io5 | Few social activity |
| Io6 | Not close to partner |
| Io7 | Low contact with child |
| Io8 | Low contact with other family members |
| Io9 | Low contact with friend |
| | |
| Is1 | Lack of companion |
| Is2 | Feel left out |
| Is3 | Feel isolated |
| Is4 | Feel alone |
| Is5 | Feel not in tune |
| Is6 | No people to talk to |
| Is7 | No people to turn to |
| Is8 | Not being understood |
| Is9 | Not part of a group |
| Is10 | Cannot find common with others |
| Is11 | No people close to |
| | |
| Fs1 | Difficulty with walking |
| Fs2 | Difficulty with dressing |
| Fs3 | Difficulty with bathing |
| Fs4 | Difficulty with eating |
| Fs5 | Difficulty with going to bed |
| Fs6 | Difficulty with using toilet |
| Fs7 | Difficulty with using a map |
| Fs8 | Difficulty with managing money |
| Fs9 | Difficulty with shopping |
| Fs10 | Difficulty with taking medication |
| Fs11 | Difficulty with making a hot meal |
| Fs12 | Difficulty with using a cellphone |
| | |
| Fo1 | Incomplete balance tests |
| Fo2 | Low grip strength |

**Fig 2. Communities identified in the network of physical functioning and social connection.** Caption for Fig 2: Colors correspond to five communities in the network: [1] Orange: Emotional loneliness; [2] Blue: Social loneliness; [3] Pink: Low social contact; [4] Green: Core functioning; and [5] Yellow: Complex functioning. Circles refer to subjective indicators (Is: Social connection/subjective, Fs: Physical functioning/subjective) and squares to objective indicators (Io: Social connection/objective, Fo: Physical functioning/objective). See eTable 2 for node definitions and operationalizations.

## Discussion

To the best of our knowledge, this study is among the first to use network analysis to investigate the interrelationships between objective and subjective measures of physical functioning and social connection. Beyond pair-wise association in conventional analysis, network analysis allowed us to systemically map the structure of relationships between the two constructs and quantify the role of individual indicators within the network. Drawing on the DPM model, the key findings from this study are four-fold: First, the network structure is primarily determined by the constructs, rather than the types of measures. Second, there is considerable diversity: within the network of two constructs, five distinct communities are identified. Third, the community composition is not dependent on the types of measures (i.e., communities included both subjective and objective indicators), emphasizing the complexity of fully capturing the social connection and physical functioning constructs. Finally, the *incomplete balance tests* (an objective indicator of physical functioning) is the key bridge



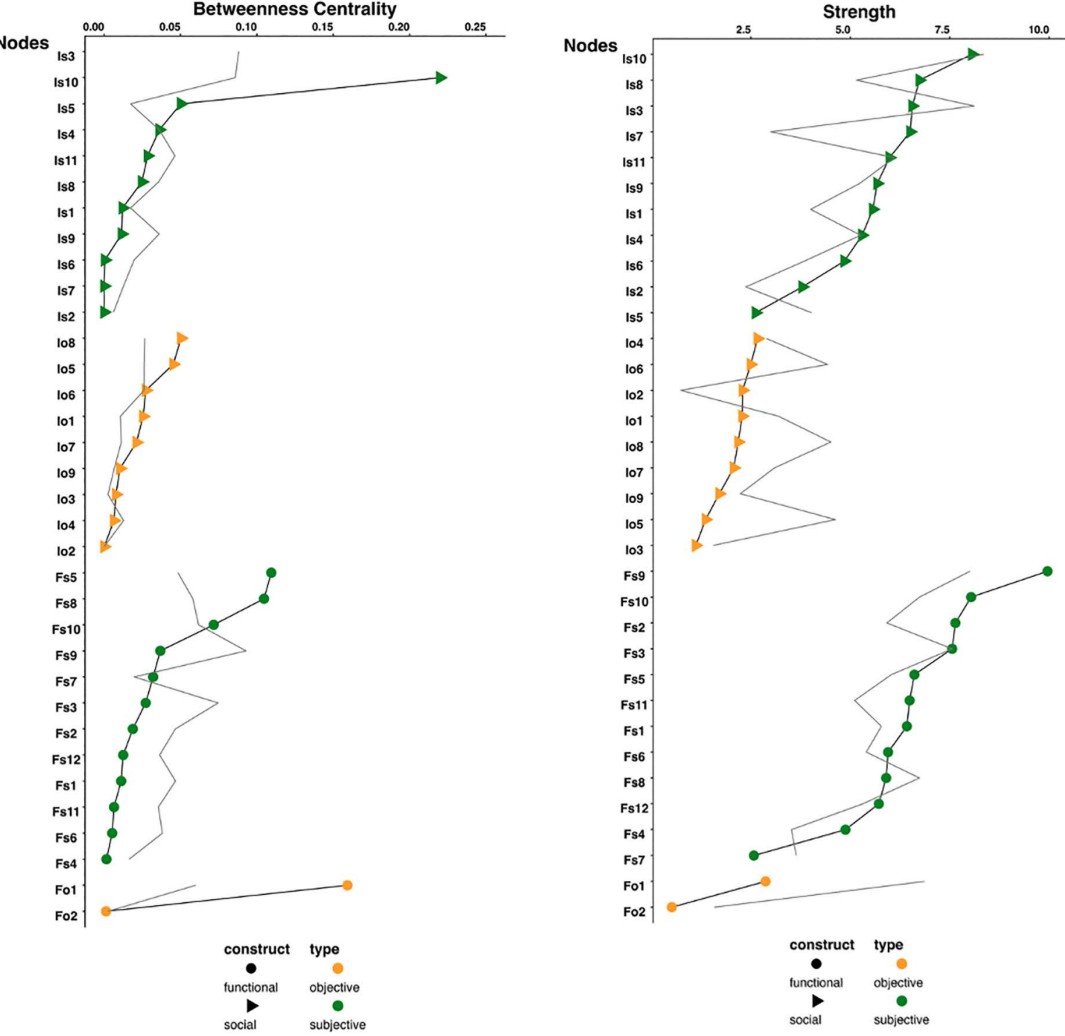

**Fig 3. Scatterplots of betweenness centrality and strength of the indicators of physical functioning and social connection. Fig 3a. Betweenness Centrality Fig 3b. Strength.** Caption for Fig 3: Green nodes refer to subjective indicators (Is: Social Connection/subjective, Fs: Physical Functioning/subjective), and maize nodes refer to objective indicators (Io: Social Connection/objective, Fo: Physical Functioning/objective). Circles refer to Physical functioning, and triangles refer to social connection. See S2 Table for node definitions and operationalizations. **Fig 3a** showed the ranking of betweenness centrality values of 34 nodes. **Fig 3b** showed the ranking of strength values of 34 nodes. The gray lines indicate the mean parameter for each node estimated from 100 null models.

that connects the associations between the two constructs, physical functioning and social connection. The implications of these findings are discussed in turn below.

The result of community detection is meaningful in two different ways. First, the community composition aligns with the theoretically-driven subgroups that are not pre-defined in the network. For example, one identified community captures seven subjective indicators (e.g., I can turn to someone, I feel close to someone). Those seven subjective indicators are the same indicators of social loneliness, a subgroup within the UCLA loneliness scale, referring to a lack of membership in a desired group [37,38], which we did not specify in the network. Second, subjective and objective indicators often share community membership. Specifically, in the social loneliness community, the seven subjective indicators share membership with one objective indicator (i.e., I have no friends). It is worth noting that this "internal structure" of the association

is not pre-defined but is detected from the network. This information is not accessible to other approaches for examining interrelated relationships, such as confirmatory factor analysis, which is better suited for testing a priori specified hypotheses. The detected communities emphasize that although social isolation (objective) and loneliness (subjective) are conceptually distinct, they remain correlated [39]. Taken together, intervention developers or practitioners should consider both aspects of social connection. A meta-analysis found that purely social interventions showed limited effect in reducing loneliness [40]. One possible explanation was that prescribed social interventions may not readily foster meaningful relationships [40]. Future peer-befriending programs should consider longer duration or incorporate opportunities for deeper conversations, enabling participants to not only connect with peers (objective goal) but also potentially foster trust and closeness(subjective goal).

The *incomplete balance tests* is the key bridge of social connection and physical functioning is an indicator-level granularity that the network analysis offered. It has important implications in research and practice. Theoretically, it extends the DPM by highlighting the potential pathways linking physical and social aspects of DPM. Given that balance is a complex feature that is influenced by multiple systems such as muscle strength, posture control, and vision [41], the relationship between balance impairment and social connection is complex, with multiple phases. Initially, the association may primarily be unidirectional, where balance impairment is associated with the incidence of falls, as well as fear of falling [42]. These experiences may reduce time and opportunity to leave the home, thereby increasing social isolation and loneliness [43,44]. In the later phase, the association may become bidirectional, as prolonged reduction in social connection contributes to less physical activity, which further diminishes lower-body strength and worsens balance impairment [45].

Practically, this finding provided empirical evidence for multimodal interventions aiming at improving physical functioning among older adults. For example, Experience Corps ® is a community-based program designed for older adults. It first provides older adults with training in exercising, social, and cognitive skills with peers, and then they volunteer as tutors for elementary school students [46]. This program has shown significant improvement in physical functioning, social engagement, and cognitive functioning [46]. Examples of physical activity interventions include SilverSneakers ®, a fitness program that provides free gym membership and access to exercise classes for eligible Medicare beneficiaries [47], and Leverage Exercise to Aging in Place (LEAP), a virtual exercise program that brings older adults together for workout partners [48]; these programs not only encourage engagement in physical activity but also create a sense of community (e.g., through exercise partnership) and opportunities build connection with other older adults.

Collectively, our findings illustrate that network analysis, as a novel approach, enables us to explore and visualize the structure of complex associations between social connection and physical functioning, offering unique opportunities to generate hypotheses for future examination.

## Strengths and limitations

The study has several strengths. First, it demonstrates a novel application of network analysis to examine the relationship between physical functioning and social connection. This approach allowed us to characterize the associations between these two constructs in a more nuanced manner. Second, we conducted several robustness checks, including ablation tests, assessment of the network structure under different assumptions, and comparison to a null network, which increases the rigor of our analysis. Third, we investigated the associations within and between various objective and subjective indicators of physical functioning and social connection simultaneously rather than as aggregated indices. Finally, the HRS is one of the largest, most well-characterized samples of older adults, which enhances the external validity of our inferences.

Regarding limitations, we identified the following. First, this is a cross-sectional analysis using pooled data from the 2014 and 2016 interview waves of the HRS and, therefore, limits our ability to infer causation from the findings. Second, this analysis used the 2014/2016 interviews instead of the most recent waves of HRS because of a skip pattern error in the 2018 wave. As a result, the responses to functional limitation questions are unreliable for ADL assessment [25], and

the 2020 wave was impacted by the pandemic. Future studies should investigate how pandemic-related policies (e.g., social distancing, nursing home closures) may have affected the relationship between physical functioning and social connection. Third, we acknowledge that dichotomization is not required in network analysis, and we lost variance of indicators by dichotomizing all indicators. Fourth, the HRS has a limited number of objective measures of physical functioning (e.g., balance test and grip strength) due to costs and time constraints. The number of indicators is not balanced between types of measures in physical functioning (2 objective indicators vs. 12 subjective indicators). Future studies can include additional objective measures of physical function to examine these associations in a more comprehensive manner. Finally, the analytic sample was more likely to be younger, non-Hispanic White, and to have higher socioeconomic status. Consequently, the finding may be subject to selection bias, and the observed association has limited generalizability to minority populations and individuals with disadvantaged socioeconomic status.

## Conclusion

This cross-sectional study of 7,270 older US adults used network analysis to visualize the complex association between physical functioning and social connection. This analysis detected five communities from the network of two constructs and identified that the objective indicator of physical functioning (i.e., balance) is the key bridging node linking these two constructs. Aligning with the Disablement Process Model, these findings emphasize that interventions aimed at promoting physical functioning for older adults would benefit from also incorporating elements that promote social connection (i.e., multimodal programs such as Experience Corps ®).

## Supporting information

**S1 Table. Demographic Characteristics of the analytic sample and the excluded sample.**
(ZIP)

## Author contributions

**Conceptualization:** Chuwen Zhong, Linh Dang, Aparna Ananthasubramaniam, Briana Mezuk.

**Data curation:** Chuwen Zhong, Linh Dang.

**Formal analysis:** Chuwen Zhong, Aparna Ananthasubramaniam.

**Funding acquisition:** Briana Mezuk.

**Methodology:** Aparna Ananthasubramaniam.

**Project administration:** Briana Mezuk.

**Resources:** Linh Dang, Briana Mezuk.

**Software:** Aparna Ananthasubramaniam.

**Supervision:** Briana Mezuk.

**Visualization:** Chuwen Zhong.

**Writing – original draft:** Chuwen Zhong.

**Writing – review & editing:** Chuwen Zhong, Linh Dang, Aparna Ananthasubramaniam, Briana Mezuk.

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
