## [Decision Letter · Decision Letter 0]

27 Aug 2025

Dear Dr. Zhong,

Thank you for submitting your manuscript to PLOS ONE. After careful consideration, we feel that it has merit but does not fully meet PLOS ONE’s publication criteria as it currently stands. Therefore, we invite you to submit a revised version of the manuscript that addresses the points raised during the review process.

We look forward to receiving your revised manuscript.

Kind regards,

Cheong Yu Stephen Chan

Academic Editor

PLOS ONE

Journal Requirements:

“This work was supported by the National Institute of Health/National Institute of Mental Health (R01 MH128198).”

3. Thank you for uploading your study's underlying data set. Unfortunately, the repository you have noted in your Data Availability statement does not qualify as an acceptable data repository according to PLOS's standards.

4. Please include captions for your Supporting Information files at the end of your manuscript, and update any in-text citations to match accordingly. Please see our Supporting Information guidelines for more information: http://journals.plos.org/plosone/s/supporting-information .

Additional Editor Comments:

**I carefully read both reviewers' comments, you are suggested to improve your manuscript in the following domains:**
**-**
**-**

**Comments to the Author**

1. Is the manuscript technically sound, and do the data support the conclusions?

Reviewer #1: Partly

Reviewer #2: Partly

2. Has the statistical analysis been performed appropriately and rigorously?

Reviewer #1: I Don't Know

Reviewer #2: Yes

3. Have the authors made all data underlying the findings in their manuscript fully available?

Reviewer #1: Yes

Reviewer #2: Yes

4. Is the manuscript presented in an intelligible fashion and written in standard English?

Reviewer #1: No

Reviewer #2: No

Reviewer #1: The researchers claim to have developed a new approach, namely to measure the relationships between different concepts with the help of network analysis. I found it interesting to become acquainted with this approach. My opinion is that the network analysis can be a valuable addition to existing possibilities, but that this paper still has too many flaws to be considered for publication at this time. (Theoretically, I am familiar with network concepts, but my expertise with applied network analysis is limited.)

Comments and suggestions

The article focuses on the link between social connection and functioning. However, functioning can be in all kinds of areas (including social); so it must be about physical functioning. Even then, the two terms are not completely valid. Loneliness is much more than social functioning, and also involves emotional functioning, for example. ADL and IADL are also more than just physical functioning, as is rightly argued in the article. The title of the article is therefore unclear and invalid.

The introduction discusses the necessity of applying a network analysis. However, this is not convincing. For example, “empirical understanding of the linkages between these constructs is hampered by differences between ‘objective’ (e.g., grip strength, network size) and ‘subjective’ (e.g., activities of daily living, loneliness) measurement schemes”. What is the objection here? Isn't it strange that the applied network analysis leads equally well to a distinction parallel to objective and subjective concepts? The second aspect, namely that other research ignores the multi-dimensional structure of concepts, is not convincing either. Contrary to what is stated on page 7, there is in fact a lot of research in which constructs in the two domains of functioning are multidimensional. It is noteworthy that the researchers themselves use the UCLA scale, which was developed as a unidimensional scale, while there are also various multidimensional loneliness instruments. The researchers also object to summary scores, but they use them themselves (Table S2). It is not clear to me why network analysis is absolutely necessary, and why a confirmatory factor model has not been developed in which latent concepts are measured by different indicators, with relationships between the latent concepts. On page 6, the researchers again express objections to an existing technique, this time to regression-based approaches. Unfortunately, they do not explain why these are unsuitable or why they do not lead to improved interventions. The researchers should better substantiate their criticism of other methods and not selectively cherry-pick from those other methods. It would also be preferable to discuss this in one section rather than scattered throughout the paper.

The researchers see their measurements of ADL and IADL functioning as subjective. I do not share this opinion. This is about objective functioning, as assessed by survey questions. I am familiar with another study using a more extensive set of performance measurements and including observations of a number of ADL and IADL functions. This indicates that the phenomenon itself can be observed objectively – of course, bias can arise when questions are asked in a survey.

I also find it unbalanced to use two performance measurements and twelve survey questions. The fact that twelve questions are needed to measure ADL and IADL functioning already indicates that the content of these concepts is much broader than being able to balance and grip strength. This requires a much more precise conceptualization of physical functioning.

The researchers distinguish two components of social connection, namely structural characteristics and perceived quality. However, the literature is more accurate than what is currently presented. There are (at least) structural and functional (e.g. social support and care) characteristics, and the perceived quality of both can be determined. These are different theoretical concepts (or constructs – that is a synonym), and it is incorrect to interpret the differences as merely a difference in measurement (page 6, line 125).

The selection in Figure S1 is very crudely displayed. What is the consequence of the selective dropout?

Why have all indicators been dichotomized (page 8)? If this is necessary for the technique, does not the removal of much variance then present an objection to the choice of network analysis?

The UCLA scale has twenty items, not eleven (page 9). Why this selection, and on what grounds were these eleven items selected?

What is the reliability and validity of the Social Disconnectedness Scale?

What technique was used to adjust for demographic covariates in the relationships between the 34 indicators? Why was the measurement level of these covariates lowered?

If less than 10% of respondents had any ADL or IADL limitations, is this sufficient to be able to carry out the analyses?

The representation in a two-dimensional space was done with the spin glass community detection algorithm – it is unclear to me what the length of the lines means and to what extent figure 2 differs from figure 1. The display reminded me of the placement of items or other units in a two-dimensional space as done in an old program called MINISSA (https://www.newmdsx.com/MINISSA/minissa.htm) based on Euclidean distance. To what extent do the results of the network analysis differ from such a graphical representation?

The accessibility of the results would benefit from a clear explanation of what they mean in terms of content.

The conclusion is drawn that “that falling the balance test played the most important role in linking functioning and social connection” (page 18). (Should it be "failing"?) But looking at figures 1 and 2, I do not see that. On what is this conclusion based, and why is this not visible in the figures?

What does the conclusion “Collectively, these findings suggest that functioning and social connection have a multifaceted association that reflects both performance-based of functioning and perceived aspect of social connection” add to existing knowledge in this domain?

Why is it possible to use this study based on data from older adults to make suggestions for interventions for “struggling students in public elementary schools”?

Quality of writing. There are many errors in spelling or in sentence structure. This indicates sloppiness in dealing with texts and possibly also in dealing with data.

References. This needs to be improved. I will give a few examples. A sentence such as “For example, older adults with functional limitations are more likely to experience shrinkage of life-space and decreased social activities.” needs elaboration and more references. No reference is given for the definition of loneliness (it says: “(reference)”). On page 4, lines 88 and 89, sources are mentioned, but these are not good references. On what do the authors base the conclusion that “the precise mechanisms underlying these relationships are unclear” (page 4)? The association between social connection and health outcomes has been extensively researched, and fortunately much has also been done to work out the underlying theoretical mechanism. On page 5 it is stated: “the empirical research on this relationship is mixed” (I think this must refer to the results of the research). Not only is this statement contrary to the conclusion that the relationship is unclear, but it is also not true that the results are mixed. In the example mentioned, in which ELSA and HRS are compared, it concerns two very different outcomes, probably with different underlying mechanisms as well. The description of the Disablement Process Model mentions “intra-individual factors (e.g., loneliness)”, but loneliness is not mentioned in either of the two referenced publications.

Ethics of the research. I see no reference to a publicly available report on the processing of the original data.

Reviewer #2: 1. Novelty:

The study presents an innovative approach by examining both subjective and objective indicators of social connection and functioning using network analysis, moving beyond traditional regression methods. The identification of balance impairment and loneliness as key bridging nodes between functioning and social connection provides valuable, actionable targets for intervention and policy development. The manuscript would benefit from a more in-depth exploration of the potential mechanisms underlying the strong association between isolation and imbalance. Further elaboration on the real-world applicability of identifying the bride node and their capacity to inform targeted intervention strategies is advised. Explicating how these findings can be translated into concrete, actionable recommendations would greatly enhance the paper’s value for both practitioners and policymakers

2. Conceptual Framework & Operationalization:

The study is well-grounded in a sound theoretical framework linking physiological impairments with the social environment, and the research question is clearly situated within existing literature. The distinction and precise operationalization of objective versus subjective indicators strengthen the analytical rigor and provide nuanced insight. However, the framework would benefit from clarification on the rationale for classifying perceived loneliness as an objective indicator, as this appears counterintuitive.

3. Generalizability

The exclusion of participants who were more socioeconomically disadvantaged and from minority groups may introduce bias into the study. It may be helpful for the authors to explain the rationales, reflect on the potential impact of this exclusion on their findings, or recognize it among the study’s limitations

4. Measurement Dichotomization:

The dichotomization of continuous or ordinal variables (e.g., grip strength) may reduce variability and sensitivity, potentially limiting the precision of the analysis. The authors are encouraged to provide a rationale for this approach or consider alternative analyses utilizing raw or more finely graded data.

5. Measurement Selection

To strengthen the methodological rigor, it would be beneficial for the authors to provide a justification for the selection of these specific measures by referencing their theoretical relevance or reporting on established psychometric properties (e.g., reliability or validity from previous empirical studies). Providing such rationales would enhance the interpretability of the findings.

6. Rigor in Analysis:

The application of appropriate network estimation techniques, comprehensive stability assessments, and adjustment for relevant covariates collectively strengthens the validity and reliability of the findings. Although sensitivity analyses for γ were conducted, a clearer rationale for choosing the default value of 0.25 and an interpretation of what increasing or decreasing γ entails for network sparsity would strengthen the methodological transparency.

7. Community Interpretation:

The manuscript’s impact could be enhanced by offering a more in-depth interpretation of the identified communities, including potential underlying mechanisms, theoretical implications, and alignment with prior research. Additionally, discussing the relevance of these communities for designing targeted interventions would add practical significance.

8. Proofreading

You are suggested to proofread the word in detail (e.g., spelling, grammar, spacing, the statistic in tables, etc)

**Do you want your identity to be public for this peer review?** For information about this choice, including consent withdrawal, please see our Privacy Policy

Reviewer #1: No

Reviewer #2: No

---

## [Author Response · Author response to Decision Letter 1]

27 Oct 2025

A detailed response letter has been uploaded as a separate file. Thanks to the reviewers for their attention and time invested in this manuscript.

---

## [Decision Letter · Decision Letter 1]

25 Nov 2025

Dear Dr. Zhong,

Thank you for submitting your manuscript to PLOS ONE. After careful consideration, we feel that it has merit but does not fully meet PLOS ONE’s publication criteria as it currently stands. Therefore, we invite you to submit a revised version of the manuscript that addresses the points raised during the review process.

We look forward to receiving your revised manuscript.

Kind regards,

Cheong Yu Stephen Chan

Academic Editor

PLOS ONE

Journal Requirements:

Additional Editor Comments:

Please revise based on reviewer's comment, enhance the clarity and language of the manuscript.

Reviewers' comments:

Reviewer's Responses to Questions

**Comments to the Author**

Reviewer #2: (No Response)

2. Is the manuscript technically sound, and do the data support the conclusions?

Reviewer #2: Yes

3. Has the statistical analysis been performed appropriately and rigorously?

Reviewer #2: Yes

4. Have the authors made all data underlying the findings in their manuscript fully available?

Reviewer #2: No

5. Is the manuscript presented in an intelligible fashion and written in standard English?

Reviewer #2: Yes

Reviewer #2: Review Comments

• The study demonstrates methodological rigor by employing network analysis alongside additional analyses (e.g., bootstrapping, ablation testing, null model comparison), enhancing confidence in the robustness of results.

• Multiple measurement modalities (objective and subjective metrics for both constructs) are presented, allowing nuanced assessment and triangulation of results.

• The use of network analysis in this domain is innovative and provides added value by identifying bridging indicators that inform targeted, multi-modal interventions for older adults.

• The manuscript’s conceptualization of “subjective” measurement is not fully resolved. It would enhance clarity to directly discuss whether perceived difficulty is equivalent to subjectivity and to distinguish environmental influence (e.g., adaptive home environments reducing perceived difficulties) from inherently subjective self-perception. This distinction is important for interpreting responses to ADL/IADL questions and understanding measurement validity.

• The rationale for choosing psychometric network analysis over conventional regression-based approaches remains insufficiently developed. It is recommended to explicitly state what methodological gaps or limitations in prior literature necessitate network analysis for these constructs, beyond precedent in mental health symptom comorbidity research.

• Clarify the unique insights network analysis can provide regarding the multidimensional relations between physical functioning and social connection, such as detecting communities, examining individual indicator centrality, or visualizing complex patterns that are not accessible using aggregated regression models.

• The statement that excluded participants were more likely female, racial/ethnic minorities, or of lower socioeconomic status warrants further elaboration. It is advised to provide a transparent rationale for these exclusions, as it has implications for the representativeness and generalizability of the findings.

• While the manuscript defines network indices (strength and betweenness), it insufficiently addresses their selection and utility. You may consider justify why these specific indices were chosen and discuss how findings from strength vs. betweenness analyses advance theoretical or practical understanding in this context.

• Greater attention is needed to grammatical accuracy and tense usage, particularly in sections describing data collection (should use past tense).

**Do you want your identity to be public for this peer review?** For information about this choice, including consent withdrawal, please see our Privacy Policy

Reviewer #2: No

---

## [Author Response · Author response to Decision Letter 2]

3 Jan 2026

Our response to reviewers is submitted as a separate document. Thank you for the comments and opportunity.

---

## [Decision Letter · Decision Letter 2]

27 Jan 2026

Exploring the Links Between Social Connection and Physical Functioning Among Older Adults:  A Network Analysis

PONE-D-25-04189R2

Dear Dr. Zhong,

We’re pleased to inform you that your manuscript has been judged scientifically suitable for publication and will be formally accepted for publication once it meets all outstanding technical requirements.

Kind regards,

Cheong Yu Stephen Chan, Ph.D.

Academic Editor

PLOS One

Additional Editor Comments (optional):

Reviewers' comments:

Reviewer's Responses to Questions

**Comments to the Author**

Reviewer #2: All comments have been addressed

2. Is the manuscript technically sound, and do the data support the conclusions?

Reviewer #2: Yes

3. Has the statistical analysis been performed appropriately and rigorously?

Reviewer #2: Yes

4. Have the authors made all data underlying the findings in their manuscript fully available?

Reviewer #2: Yes

5. Is the manuscript presented in an intelligible fashion and written in standard English?

Reviewer #2: Yes

Reviewer #2: The author addressed all the comments. The manuscript looks good in terms of theoretical background, analysis, interpretation, ready to proceed.

**Do you want your identity to be public for this peer review?** For information about this choice, including consent withdrawal, please see our Privacy Policy

Reviewer #2: No

---

## [Editor Report · Acceptance letter]

PONE-D-25-04189R2

PLOS One

Dear Dr. Zhong,

I'm pleased to inform you that your manuscript has been deemed suitable for publication in PLOS One. Congratulations! Your manuscript is now being handed over to our production team.

Kind regards,

on behalf of

Dr. Cheong Yu Stephen Chan

Academic Editor

PLOS One